# Compressive Behavior of Bamboo Sheet Twining Tube-Confined Concrete Columns

**DOI:** 10.3390/polym13234124

**Published:** 2021-11-26

**Authors:** Xunyu Cheng, Yang Wei, Yuhan Nie, Gaofei Wang, Guofen Li

**Affiliations:** 1College of Civil Engineering, Nanjing Forestry University, Nanjing 210037, China; cxyyhjs@163.com (X.C.); nyh199607@163.com (Y.N.); wgf19970925@163.com (G.W.); lgf@njfu.edu.cn (G.L.); 2Jiangsu Expressway Engineering Maintenance Technology Co., Ltd., Nanjing 211106, China

**Keywords:** bamboo sheet, bamboo composite tube, confined concrete, lateral-to-axial strain relationship, ultimate strength

## Abstract

This study experimentally investigated various axial compressive parameters of a new type of confined concrete, which is termed bamboo sheet twining tube-confined concrete (BSTCC). This new composite structure was composed of an outer bamboo composite tube (BCT) jacket and a concrete core. Under axial compression, the parameters of thirty-six specimens include concrete strength (i.e., C30 and C50) and BCT thickness (i.e., 6, 12, 18, 24, and 30 layers). The mechanical properties of the BSTCC specimens from the perspective of the failure mode, stress-strain relationship, effect of BCT thickness and dilation behavior were analyzed. The results showed that, in compression, with an increase in BCT thickness in the range of 18-layers of bamboo sheets, the strength increased remarkably. When the strength of the concrete core was high, the confinement effect of the BCT was reduced. In addition, the BCT thickness relieved the dilation of the BSTCC specimens. Finally, the experimental results were compared with predictions obtained from 7 existing FRP-confined concrete models. All the predictions had good agreement with the test results, which further confirmed that the models developed for FRP-confined concrete can provide an acceptable approximation of the ultimate strength of the BSTCC specimens.

## 1. Introduction

It is well known that steel tubes and fiber reinforced polymer (FRP) material as external confinement of concrete enhances concrete axial compressive strength and ductility significantly [1,2,3]. These FRP tubes or steel tubes, acting as stay-in-place structural forms for fresh concrete, can also protect the encased concrete from potentially aggressive environments, e.g., deicing salts and other chemicals [4,5,6]. However, with the growing awareness of environmental protection and the emergence of problems, such as the energy crisis and carbon emissions, the use of natural materials instead of petroleum-based materials has gradually become a research hotspot and has received widespread attention [7].

In recent years, natural fiber reinforcements, as alternatives to carbon or aramid fibers, have been widely applied in aerospace, automobile, infrastructure, and other industrial fields [8,9]. The advantages of natural fibers over synthetic fibers, such as fiberglass and carbon fibers, are low cost, low density, good tensile properties, and reduced energy consumption, renewability, and biodegradability [10,11]. At present, the use of natural fiber composites in civil engineering is rare, especially for the use of natural fiber composites as external strengthening materials. It is of great interest that flax fibers have been developed to reinforce concrete columns and concrete beams. Yan et al. [12] conducted a series of studies on the mechanical properties of concrete confined by flax FRP tubes. Xia et al. [13] tested the axial compression properties of bidirectional flax fiber reinforced concrete cylinders and obtained the performance characteristics of flax fabric reinforced polymer (FFRP) reinforced concrete by comparison with existing models. Sen et al. [14] conducted an experimental investigation on the confinement strength and confinement modulus of concrete cylinders confined using different types of natural fiber composites and made a comparative performance analysis with different artificial fiber-based composite materials.

According to previous studies, bamboo is a very attractive material due to its renewable short natural growth cycle and abundance of bamboo resources [15,16,17,18,19]. Bamboo can be used as columns, beams, and floors, alone or combined with wood, steel, and concrete, to form composite components to give full play to the advantages of the various materials [20,21,22,23,24,25,26,27]. In addition, bamboo fiber can offer excellent mechanical properties and is common used as reinforcement in engineering applications, which has attracted attention over other natural fibers. Chiu et al. [28], Sanchez et al. [29], and Chin et al. [30] introduced the mechanical properties of bamboo fiber reinforced composites. The results showed that the treated bamboo fiber could effectively improve the strength of the biocomposite materials. Javadian et al. [31] reported a use of bamboo fiber-reinforced polymer composite as reinforcement for structural-concrete beams. The results indicated that there was significant potential for practical implementation of the bamboo-composite reinforcement.

If the bamboo is wound to form a tube and the core is filled with concrete, the excellent tensile performance of bamboo will be expected to develop to confine the concrete core. Therefore, to verify the feasibility of using bamboo composite tubes as external confining materials for a concrete core, Kou et al. [32] proposed a new method of bamboo use, which is to twine laminated bamboo lumber slices to form a bamboo-fiber-reinforced composite tube. The results indicated that, for structural application, natural bamboo fiber reinforced composite tubes had favorable confinement capabilities.

On the base of the structure of bamboo composite tube confined concrete, the mechanical behavior of bamboo sheet twining tube-confined concrete (BSTCC) under axial compression was investigated. The bamboo was wound to form different thick composite tubes, and concrete filled its core. The influence of the wall thickness of bamboo composite tube and different concrete strength on the confinement of the concrete core were fully analyzed. Parameters of the specimens in this paper vary over wide ranges, with unconfined concrete strength from 39.15–70.59 MPa and bamboo sheet layers (6–30). In addition, the new tests significantly widened the range of confinement ratios (the ratio of the ultimate confining pressure to the unconfined concrete strength) of 0.04–0.34 for the 36 tests considered in this paper. Utilizing the good tensile strength of bamboo, the results presented in the study are expected to provide a new structure for confining concrete with bamboo composite tubes.

## 2. Experimental Program

### 2.1. Specimen Design

Thirty-six specimens (thirty bamboo sheet twining tube-confined concrete columns (BSTCC) and six unconfined concrete columns) with inner diameters of 150 mm and heights of 300 mm were designed and prepared for the monotonic axial compression test. The main variation parameters included the number of bamboo sheet layers (0, 6, 12, 18, 24, or 30 layers), which is the thickness of the bamboo composite tube (BCT), and the grade of the concrete (C30 or C50). The specific parameters of the specimens are shown in Table 1. Three identical specimens were made for each set of designed parameters. The specimens were named according to the grade of the concrete and the number of bamboo sheet layers. The first letter “C” and the following number represent the grade of the concrete (C30 or C50); the second letter “B” and the following number represent the number of bamboo sheet layers (B6, B12, B18, B24, or B30). “N” indicates that it is a plain concrete specimen without BCT confinement. For example, C30B6 represents BSTCC columns with C30 concrete and 6 bamboo sheet layers.

### 2.2. Materials

#### 2.2.1. Concrete

The cement used in the test was Portland 42.5 (Anhui Conch Group Co., Ltd., Wuhu, China). The specimens were cast in two batches with two different concrete mix ratios. To determine the concrete strength, three concrete cubes with dimensions of 150 × 150 × 150 mm were prepared in the same batch for each mix ratio. The specimens were cured under the same conditions for 28 days. It can be seen from Table 2 that the average compressive strength of the two batches of concrete cubes were 39.15 MPa and 70.59 MPa. The values of standard deviation (SD) and coefficient of variation (COV) is small, which means that the dispersions of the compressive strength are satisfactory.

#### 2.2.2. Bamboo Sheet

Bamboo sheets provided by Haofeng Decoration Material Co., Ltd. (Dongguan, China) were used to manufacture composite tubes as the confined material in the test. The bamboo sheets were prepared by slicing softened moso bamboo stem along the bamboo fiber, and the nominal thickness of each bamboo sheet was 0.5 mm. Five coupons of bamboo sheets from the same batch were made to obtain the tensile properties of the bamboo sheets parallel to the bamboo grain (Figure 1). The tensile strength, elastic modulus, and fracture strain values of the bamboo sheets used in the test were 76.64 MPa, 10.75 GPa, and 0.0071, respectively.

#### 2.2.3. Epoxy Resin

The epoxy resin adhesives (L-500AS/L-500BS) used in this test were provided by Sanyu Resin Co., Ltd. (Shanghai, China). The average tensile strength, elastic modulus, and ultimate tensile strain values were 67.7 MPa, 2.9 GPa, and 0.029, respectively.

### 2.3. Specimen Preparation

The detailed production steps of BCT and BSTCC are shown in Figure 2. BCTs were fabricated using the hand lay-up process in the structural engineering laboratory of Nanjing Forestry University. (1) A plastic film was applied onto the acrylic pipe before it was wrapped in bamboo sheets; thus, the acrylic pipe could be reused. (2) After saturation with epoxy resin adhesive, the prepared bamboo sheets were continuously wrapped around a horizontally placed detachable acrylic pipe with the fibers oriented solely in the circumferential direction. A 150-mm overlap length was provided for all specimens to prevent premature failure in the overlap zone. (3) Before pouring the concrete, the BCT was temporarily fixed on the bottom formwork with hot-melt glue, and the gap was sealed with glass glue to prevent leakage of the cement paste during pouring. (4) Concrete was then poured and vibrated. (5) All the specimens were cured in room conditions over 28 days. After curing, the bottom formwork was removed. The ends of all the specimens were treated with a thin coat of high-strength self-leveling mortar to ensure a uniform bearing surface. (6) The lower and upper edges of the BCT were shaped into s 45° chamfer using a polisher to prevent the BCT from bearing the axial compression.

### 2.4. Test Setup and Instrumentation

The compression tests were conducted on a high-stiffness testing machine with a capacity of 3000 kN provided by Popwil Electromechanical Control Engineering Co., Ltd. (Hangzhou, China) (Figure 3a). Prior to the test, the top and bottom surfaces were ground smooth to ensure that the load was applied uniformly across the cross-section. Two control modes were conducted during the compression tests. Each specimen was first tested using load-control at a loading rate of 10 kN/s until 10% of the pre-estimated peak load before formal loading to reduce the effect of looseness at both ends of the specimen. Then, the load control was changed to displacement control at a constant rate of 0.1 mm/min until reaching specimen failure. The loading process ended when the average axial strain reached 0.01 for BSTCC specimens. A TDS-530 data acquisition machine was used to collect the test data, including the strain and displacement at the same frequency.

The measurement of strain distribution along specimen height and perimeter may be enabled by the digital image correlation (DIC) measurement technique. However, the data processing of DIC is complex. Installing strain gauges around the mid-height section and laser displacement sensors were adopted in this test, which have been also verified in most previous studies on FRP-confined concrete [1,2,3,4,5,6]. Figure 3b illustrates the layout of the strain gauges and the linear variable displacement transducers (LVDTs). For each specimen, four strain gauges (LF1–LF4) with gauge lengths of 50 mm were installed on the specimen at mid-height 90° apart to measure the lateral strain, and another four strain gauges (AF1–AF4) with gauge lengths of 100 mm were installed at mid-height 90° apart to measure the axial strains. In addition, axial strain was also measured by two linear variable displacement transducers (LVDTs) 180° apart and covering the full height region for both the unconfined and confined specimens. Furthermore, on the other two sides, two laser displacement sensors (JD1/JD2) were symmetrically installed onto two pairs of aluminum alloy supports that bonded to the surface of the specimens, and a laser displacement transducer was fixed on the top support and aimed at the bottom support to measure the displacement of the specimen in the range of 240 mm along the specimen height.

## 3. Experimental Results

Table 3 summarizes the key test results. *P_max_* is the maximum axial load, *f_cc_* and *ε_cc_* are the peak stress and the peak strain, respectively, and *f_cu_* and *ε_cu_* denote the ultimate strength and the corresponding ultimate strain, respectively. Similar to conventional FRP-confined concrete, the confinement action of BCTs in concrete cores is passive and develops through the restraint of the lateral expansion of the concrete under uniaxial compressive loading. It is commonly assumed that the BCT ruptures when the hoop stress in the BCT reaches its tensile strength. Based on the deformation compatibility between the confining wrap and the concrete surface and assumption of a uniform confining pressure distribution, the confining pressure can be calculated using the following equation [33]:(1)$$f_{l}=\frac{2E_{f}\varepsilon_{f}t_{f}}{D}$$
where *f_l_* is the lateral confining pressure provided by the BCT. *E_f_*, *t_f_*, and *ε_f_* are the elastic modulus, thickness, and ultimate rupture strain of the BCT, respectively. *D* is the diameter of the concrete core in mm.

### 3.1. Failure Mode

The typical failure mode of the BSTCC specimens with different parameters is shown in Figure 4. All the specimens had serrated cracks throughout the axial direction of the BCT and failed by tensile rupture in the hoop direction of the BSTCC specimens. The fractured BCT then separated from the concrete core. However, the number of bamboo sheet layers had an influence on the failure pattern. When the number of bamboo sheet layers was less than 18, the fracture of the crack initiated at the middle of the BCT and then propagated towards the two ends of the specimen along its longitudinal direction resulting in the development of a major crack. When the load reaching the ultimate bearing capacity of the specimen, with a crisp sound, the mid-height region of the BCT was stripped destructively. As the number of bamboo sheet layers increased, when the load increased to 70–80% of its ultimate bearing capacity, the bearing capacity of the specimen increased more slowly, while its lateral deformation began to develop obviously. Only a slight tearing sound of bamboo rupture came from the inside initially, and no obvious change on the surface of the BCT was observed. With the increase in the load, the cracks developed from the middle section of BCT and developed dramatically. The BSTCC specimens failed by the BCT rupture at the entire longitudinal section of the specimens due to hoop tension, which led to a rapid and sudden drop in the load, accompanied by a huge and crisp sound. Then, the internal concrete completely failed, and the entire specimen was destroyed.

### 3.2. Stress-Strain Relationship

The axial stress versus axial/lateral strain curves of the unconfined concrete and the BSTCC specimens are shown in Figure 5. The average readings of the two laser displacement sensors with 240-mm gauge lengths were used to calculate the axial strain, and the lateral strains were averaged from the readings of the four lateral strain gauges installed on the surface of the BSTCC specimens. At the early loading stage, the readings from the laser displacement sensors were corrected by measurements from the axial strain gauges. The complete stress-strain curves can be divided approximately into three stages by two key points (the nonlinear transitional point and the ultimate point): the first stage is the linear elastic stage, the second stage is nonlinear transition region, and the third stage is the nonlinear stage. In the first linear elastic region, the slopes of the confined concrete with different bamboo sheet layers were similar to that of the unconfined concrete. This behavior implies that the BCT confinement had no obvious effect on the elastic modulus of the concrete core. As the applied load was low, nearly no lateral expansion occurred within the concrete core; thus, the confinement of the BCT was not activated. When the applied stress approached the ultimate strength of the unconfined concrete, the curve entered the second nonlinear transitional region where considerable microcracks propagate within the concrete core, and the lateral expansion increases considerably. At this stage, the BCT started to restrict the lateral deformation of the concrete. As the axial strain increased, the lateral deformation of the concrete core gradually increased. The third nonlinear stage is mainly dominated by the structural behavior of BCT, in which the BCT is fully activated to confine the concrete core, leading to a considerable increase in the compressive strength of the concrete when the core is subjected to triaxial compression.

### 3.3. Influence of the BCT Thickness

As shown in Figure 6, the test results of specimens with the same concrete strength (C30 or C50) but different layers of bamboo sheets were plotted together for a clear comparison. To study the impact of the BCT confinement level on the stress-strain behavior of BSTCC specimens, the test results in Figure 6 were grouped by 6-, 12-, 18-, 24-, and 30-layer bamboo sheet wraps and are shown with the unconfined counterpart as the control specimen. As expected, the BCT thickness considerably affected the stress-strain relationship of the BSTCC specimens. Thus, the ultimate strength and ultimate strain were significantly enhanced when the number of bamboo sheet layers was below 18 layers. Figure 6 shows that the number of bamboo sheet layers plays an important role in the compressive behavior of the BSTCC specimens. Compared with the unconfined concrete, BSTCC specimens had larger ultimate strength and ultimate strain values than unconfined concrete. Figure 7 shows the relationship between the number of bamboo sheet layers and the increase ratio of the ultimate stress for the BSTCC specimens, where the increase ratio of the ultimate stress is defined as the ratio of the ultimate stress of the BSTCC to that of the unconfined concrete. It can be observed that the increase ratio of the ultimate stress increases when the number of bamboo sheet layers is below 18 layers. Using the C30 specimens as an example, the increase ratios of the ultimate stress for the C30s with 6-layer, 12-layer, and 18-layer bamboo sheets are 1.19, 1.44, and 1.66, respectively, meaning that the increase ratios of the ultimate stress increase with an increase in the number of bamboo sheet layers. However, some differences are observed for the C30 specimens with 24-layer and 30-layer bamboo sheets, and the increase ratios of the stress are 1.54 and 1.68, respectively, meaning that the increase ratio of the ultimate stress for the BSTCC does not always increase with an increase in the number of bamboo sheet layers. Similarly, the same phenomenon was observed in the C50 specimens. Therefore, only when the layer number is within a certain range does the increase ratio of the ultimate stress increase as the number of bamboo sheet layers increases. In addition, when the strength of the concrete core is high, the confinement effect also decreases. From the comparison of the corresponding increase ratio of the ultimate stress, it can be seen that the improvement of the bearing capacity and deformation capacity of the BSTCC with a concrete strength of C30 is obviously higher than that of the BSTCC with a concrete strength of C50. For example, the increase ratios of the ultimate stress for the C30 specimens with 30-layer bamboo sheets and C50 specimens with 30-layer bamboo sheets are 1.68 and 1.04, respectively. Due to the brittle failure of high-strength concrete, the ultimate strain of BSTCC with a concrete strength of C50 is far less than that of BSTCC with a concrete strength of C30.

### 3.4. Dilation Behavior

The dilation behavior of the confined concrete can be effectively characterized by the lateral-to-axial strain relationship because the lateral dilation driven by the axial shortening of concrete is the cause of the confining pressure supplied by the BCT jacket, which in turn counteracts the lateral dilation. The lateral-to-axial strain curves of those specimens with the same concrete grade are shown together in Figure 8 to examine the effects of the confinement stiffness on the dilation behavior. In the initial stage of deformation, the dilation ratio of the BSTCC specimens is relatively small, while it begins to increase rapidly when the axial strain approached the strain corresponding to the peak stress of the unconfined concrete. It is evident that the BCT thickness (confinement stiffness) has a significant effect on the dilation behavior of the BSTCC specimens. For example, at a given axial strain, the 6-layer BSTCC specimens produced a larger lateral strain than those of the 12-layer BSTCC specimens, as expected. This proved that increasing the BCT stiffness relieves the dilation of the BSTCC specimens.

## 4. Analysis

### 4.1. Lateral-to-Axial Strain Relationship

The lateral-to-axial strain relationship is an important index in the passively confined model of confined concrete, which provides the response of interaction between the concrete core and the external jackets. Several lateral-to-axial strain models have been proposed by Lim and Ozbakkaloglu [33], Jiang and Teng [34], Chun and Park [35], Marques [36], and Lokuge et al. [37]. Figure 9 shows the comparison between the predicted and test lateral-to-axial strain curves for BSTCC specimens. The comparison more directly shows that these models are less accurate in predicting the axial strains for the specimens. The trends and values of the models predicted by Chun and Park, Marques, and Lokuge et al. deviate significantly from the tests. The prediction trend of the model by Lim and Ozbakkaloglu is consistent with the experimental results, but, when the concrete grade is higher (C50), the lateral strain is overestimated. The results from the Lim and Ozbakkaloglu [33] model suggest that further research adopting a more rigorous approach and using a wider range of test results is needed to develop a more generally applicable axial-to-lateral strain model. Therefore, to accurately describe the lateral-to-axial strain response of BSTCC specimens, a new model based on the Lim and Ozbakkaloglu model was proposed as follows:(2)$$\varepsilon_{c}=\frac{\varepsilon_{l}}{v_{i}{\left[1+{\left(\frac{\varepsilon_{l}}{v_{i}\varepsilon_{co}}\right)}^{n}\right]}^{\frac{1}{n}}}+0.04\varepsilon_{l}{}^{0.7}\left[1+11{\left(\frac{f_{l}}{f_{co}}\right)}^{0.7}\right]$$
(3)$$v_{i}=8\times 10^{-6}f_{co}{}^{2}+0.0002f_{co}+0.138$$
(4)$$\varepsilon_{co}=(-0.067f_{co}{}^{2}+29.9f_{co}+1083)\times 10^{-6}$$
(5)$$n=1+0.012f_{co}$$
where *ε_c_* and *ε_l_* are the axial strain and lateral strain, respectively; *f_co_* and *ε_co_* are the compressive strength of unconfined concrete in MPa and the corresponding axial strain, respectively; *f_l_* is the corresponding continuous confinement pressure for a given lateral strain; *v_i_* is the initial Poisson ratio of the concrete obtained from the unconfined concrete; and *n* is the curve shape parameter to adjust the initial transition radius of the predicted lateral-to-axial strain relationship curve.

As shown in Figure 9, the analytical curves based on the improved lateral-to-axial strain model showed good agreement with the experimental results.

### 4.2. Evaluation of the Ultimate Strength

For FRP-confined concrete, the ultimate compressive strength is one of the critical parameters. To date, many models have been developed to predict the ultimate stress for FRP-confined concrete. In this study, a total of seven strength models for circular FRP-confined concrete collected from the literature are listed in Table 4. The ultimate compressive strength results of all the BSTCC specimens are predicted using these strength models. Generally, the ultimate strength is modeled as a linear or nonlinear function of the confinement ratio. The notation and format of the original equations have been modified in most cases to ensure uniformity of notation within the current discussion. The most common form of FRP-confined concrete models can be represented by the following expression:(6)$$\frac{f_{cu}}{f_{co}}=\alpha +\beta {\left(\frac{f_{l}}{f_{co}}\right)}^{\eta }$$

The axial behavior of confined concrete was first proposed by Richart et al. [38]. The different relationships for *α*, *β*, and *η* of some models are listed in Table 4.

To evaluate the effectiveness of these existing FRP-confined concrete models for the BSTCC specimens, three statistical indices that were adapted to quantitatively assess the possible differences. These are the average value (*AV*), the standard deviation (*SD*), and coefficient of variation (*COV*), as given by Equations (7)–(9), respectively.
(7)$$AV=\sum \left(\frac{Theo.}{Expe.}\right)/n$$
(8)$$SD=\sqrt{\frac{1}{n}{\sum_{i=1}^{n}\left(\frac{Theo.}{Expe.}-AV\right)}^{2}}$$
(9)$$COV=\frac{SD}{AV}$$
where *Theo.* and *Expe.* represent the predicted and experimental values, respectively. *n* represents the total amount of data.

As indicated in Table 5 and Figure 10, most of the selected models can provide a high performance in estimating the ultimate strength of the BSTCC specimens. The analysis of the *AV*, *SD*, and *COV* indices indicates that the strength model proposed by Wu and Wei has the top performance in predicting the ultimate strength because the model has the most reference values for both types of BSTCC specimens with concrete strengths of C30 and C50, and the values of *AV*, *SD*, and *COV* are 0.96, 0.13, and 0.14, respectively. The Wei and Wu model also shows a relatively satisfying performance in predicting the ultimate strength for both types of BSTCC specimens with concrete strengths of C30 and C50. However, the prediction results float over a large range on both sides of the fitting line, and the SD value is 0.15, indicating that the dispersion of the model is large. The Youssef et al. model underestimates the ultimate strength for some specimens. In addition, the power coefficient of the confinement ratio is high, and the prediction results of the BSTCC specimens with a concrete strength of C50 are better than those of the BSTCC specimens with a concrete strength of C30. The overall prediction results of the models by Lam and Teng and Wu et al. are good. However, when the concrete strength is high (C50), the prediction values of these two ultimate strength models are larger than the experimental values. Similarly, the same phenomenon is observed in the Fahmy and Wu model, although the dispersion is low. The Samaan et al. model overestimates the ultimate strength of most BSTCC specimens owing to the overly large coefficient for the confinement effects, and the AV is 1.15. Because these models do not consider the effect of concrete strength, when the concrete strength is high, the prediction results are even larger. Therefore, the results indicate that the existing models developed for FRP-confined concrete can provide an acceptable approximation of the ultimate strength of the BSTCC specimens.

## 5. Conclusions

This study investigated the compressive behavior of bamboo sheet twining tube-confined concrete specimens (BSTCCs) through a series of tests, whose major parameters were the concrete strength and the bamboo composite tube (BCT) thickness. The test results are also compared with seven existing strength models for FRP-confined concrete to examine the applicability of such models to BSTCC. Based on the test results and analysis, the following conclusions can be drawn:(1)The failure mode of the BSTCC specimens is a serrated crack that initiates at the middle of the specimen, and then the crack propagates towards the two ends of the specimen along its longitudinal direction. The BCT provides strong lateral confinement, and the ability to bear axial loads as the BCT thickness increases, which limits the failure of the specimens and resists lateral expansion deformation.(2)The complete stress-strain curves can be divided into three stages for the BSTCC specimens under compression: the first linear elastic stage, the second nonlinear transition region, and the third nonlinear stage. Furthermore, the initial stiffness of the BSTCC specimens is independent of the BCT thickness.(3)Compared with the unconfined concrete, the effect of the BCT thickness on the ultimate stress and strain is significant. The compressive strength of the BSTCC specimens increases with increasing BCT thickness. However, the increase ratio of the ultimate stress does not always increase with an increase in the number of bamboo sheet layers; only when the layer number is in a certain range does the increase ratio of the ultimate stress increase as the number of bamboo sheet layers increases. In addition, when the strength of the concrete core is high, the confinement effect also decreases.(4)After evaluating the existing lateral-to-axial strain models, the results show that the model of Lim and Ozbakkaloglu has better performance. Based on the lateral-to-axial strain model proposed by Lim and Ozbakkaloglu, the authors modified the calculation formulas. The proposed model of the lateral-to-axial strain model shows good agreement with the experimental results.(5)The experimental results are compared with predictions of some existing FRP-confined concrete models. For the BSTCC specimens, the models proposed by Wu and Wei and Wei and Wu are superior to the other existing models because they are capable of predicting the ultimate stress with good accuracy and generality.

## Figures and Tables

**Figure 1 polymers-13-04124-f001:**
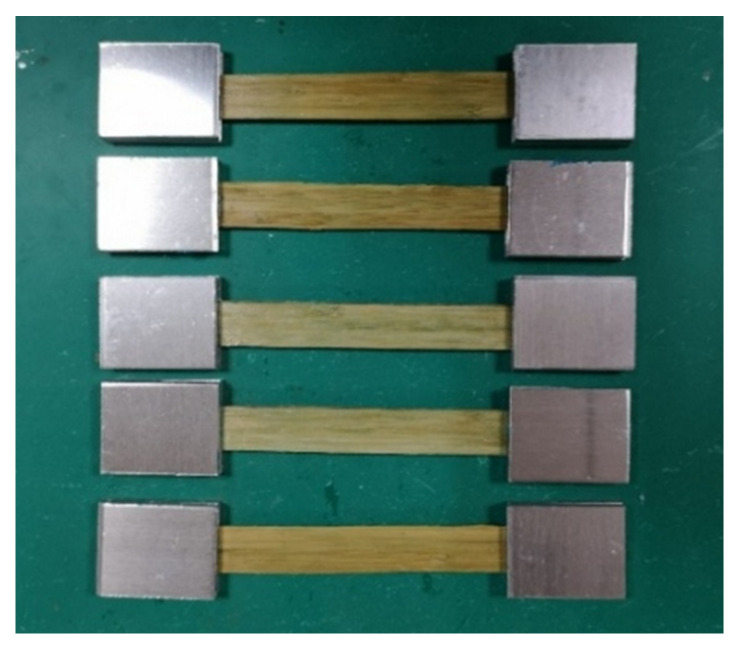
Bamboo sheet specimens.

**Figure 2 polymers-13-04124-f002:**
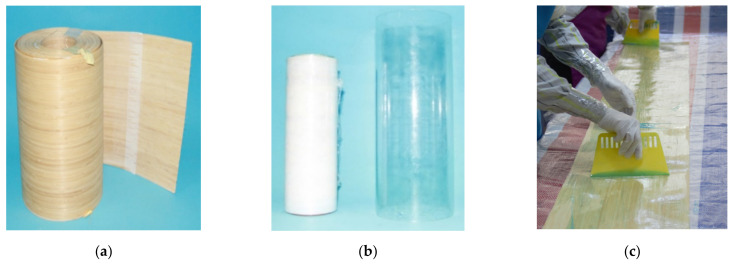
Detailed production steps of BCT and BSTCC. (**a**) Bamboo sheet; (**b**) acrylic pipe; (**c**) impregnating epoxy resin adhesive; (**d**) wrapping bamboo sheet; (**e**) BCT; (**f**) pouring concrete.

**Figure 3 polymers-13-04124-f003:**
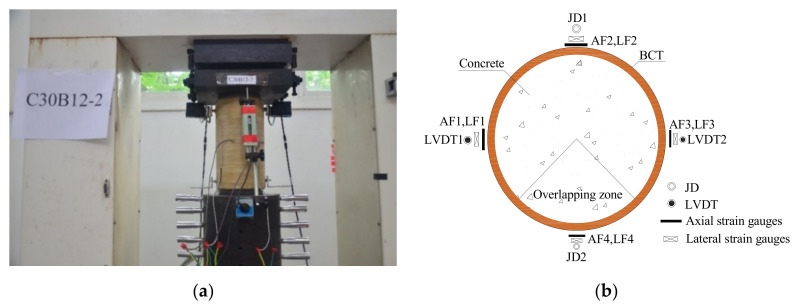
Test setup and instrumentation. (**a**) Test instrumentation; (**b**) layout of strain gauges and LVDTs.

**Figure 4 polymers-13-04124-f004:**
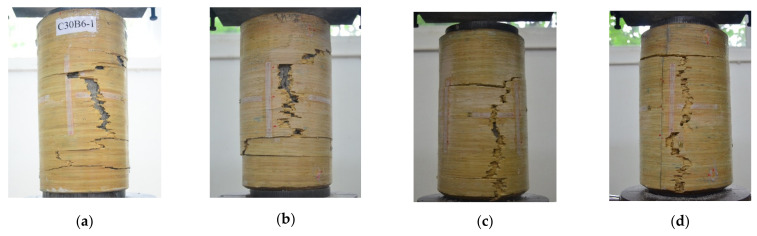
Typical failure modes of the BSTCC specimens. (**a**) C30B6-1; (**b**) C30B12-2; (**c**) C30B18-3; (**d**) C30B24-3; (**e**) C30B30-1; (**f**) C50B6-1; (**g**) C50B12-1; (**h**) C50B18-3; (**i**) C50B24-2; (**j**) C50B30-1.

**Figure 5 polymers-13-04124-f005:**
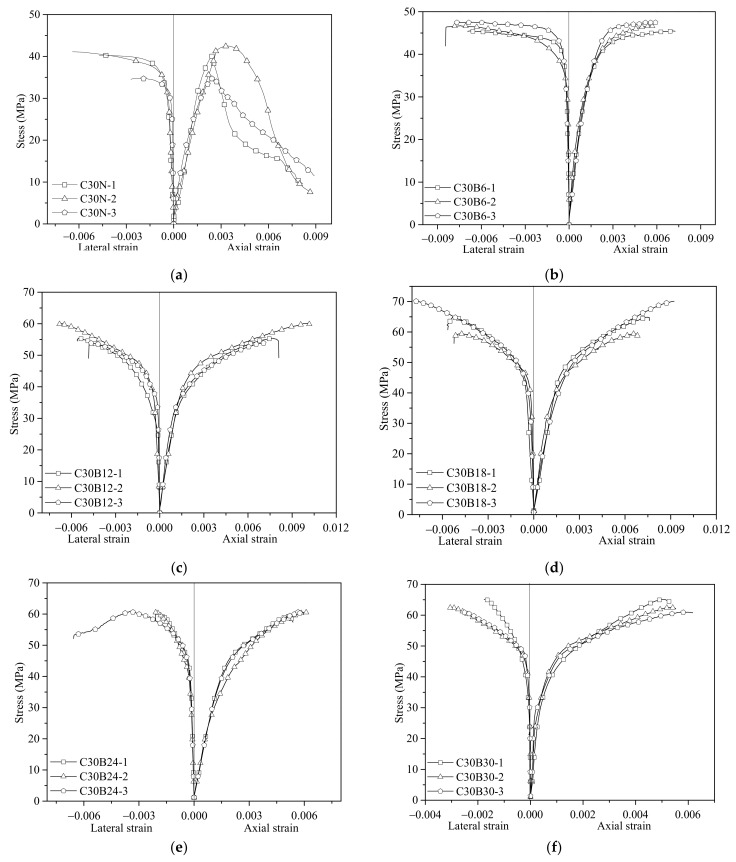
Stress-strain curves of the BSTCC specimens. (**a**) C30N; (**b**) C30B6; (**c**) C30B12; (**d**) C30B18; (**e**) C30B24; (**f**) C30B30; (**g**) C50N; (**h**) C50B6; (**i**) C50B12; (**j**) C50B18; (**k**) C50B24; (**l**) C50B30.

**Figure 6 polymers-13-04124-f006:**
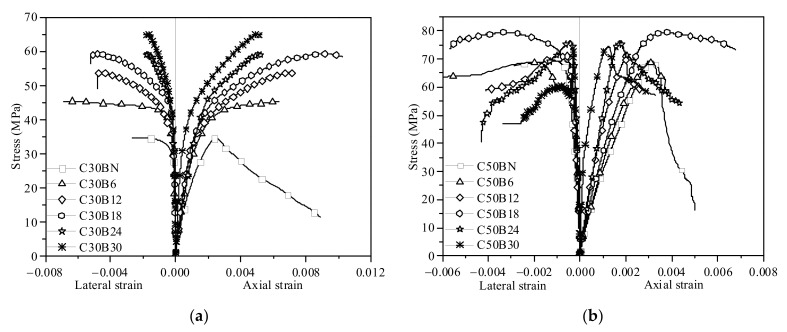
Stress-strain curves of the BSTCC specimens with the same concrete strength. (**a**) C30; (**b**) C50.

**Figure 7 polymers-13-04124-f007:**
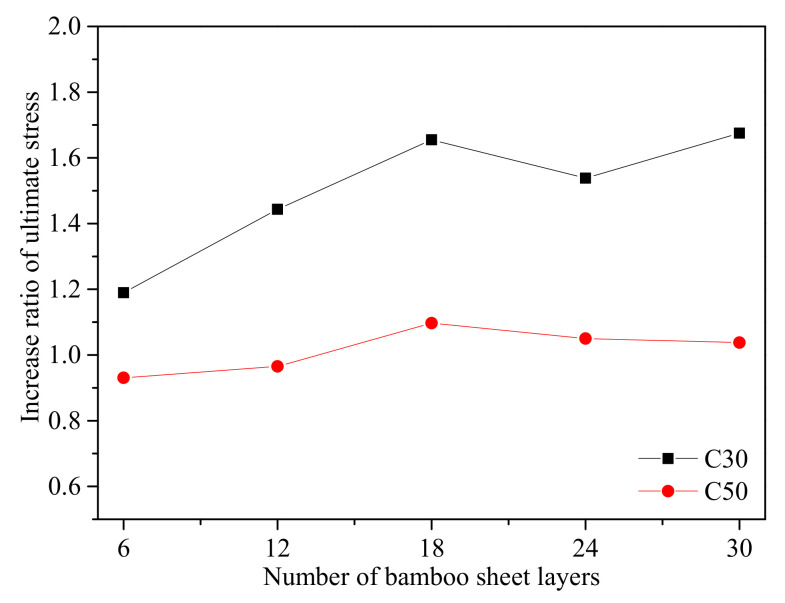
Effects of the number of bamboo sheet layers on the ultimate stress of BSTCC specimens.

**Figure 8 polymers-13-04124-f008:**
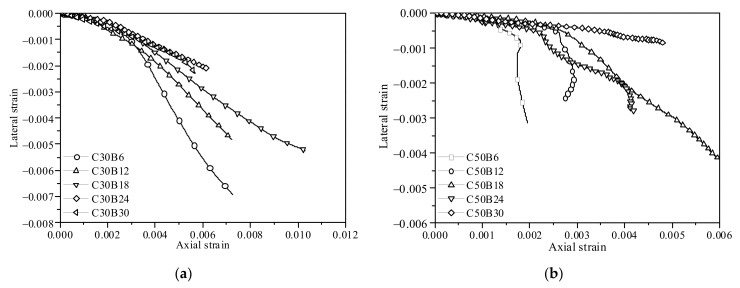
Lateral-to-axial strain curves of the BSTCC specimens. (**a**) C30; (**b**) C50.

**Figure 9 polymers-13-04124-f009:**
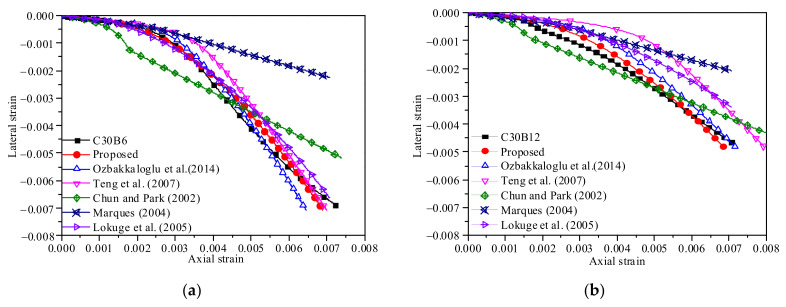
Predicted axial-to-lateral strain curves of different models for the BSTCC specimens. (**a**) C30B6; (**b**) C30B12; (**c**) C30B18; (**d**) C30B24; (**e**) C30B30; (**f**) C50B6; (**g**) C50B12; (**h**) C50B18; (**i**) C50B24; (**j**) C50B30.

**Figure 10 polymers-13-04124-f010:**
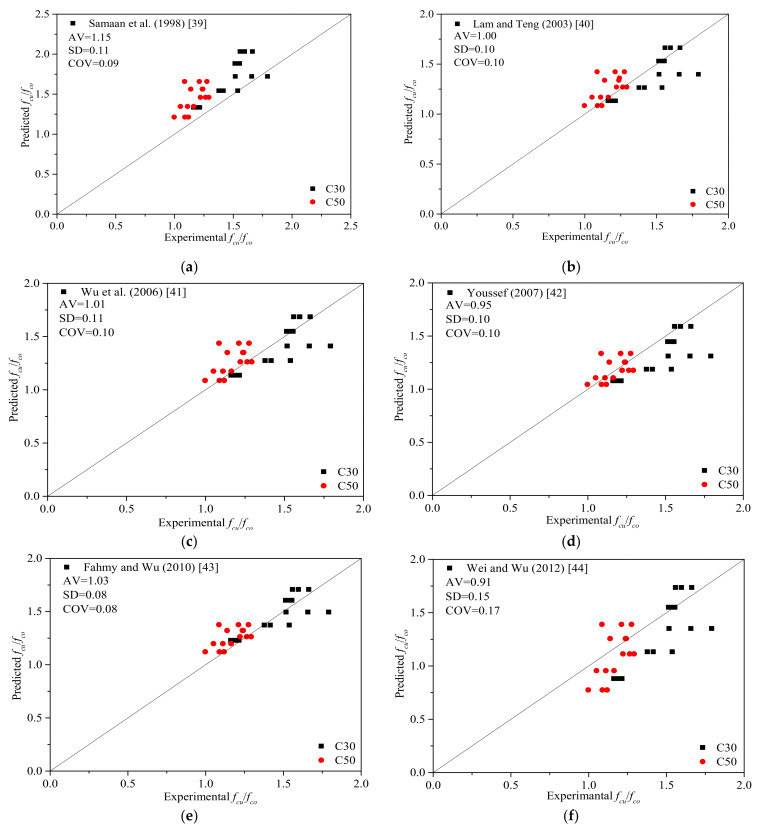
Evaluation of the calculations of the ultimate strengths of the BSTCC specimens. (**a**) Samaan et al. [39] model; (**b**) Lam and Teng [40] model; (**c**) Wu et al. [41] model; (**d**) Youssef et al. [42] model; (**e**) Fahmy and Wu [43] model; (**f**) Wei and Wu [44] model; (**g**) Wu and Wei [45] model.

**Table 1 polymers-13-04124-t001:** Specimen parameters.

No.	Specimen	Concrete Grade	Bamboo Sheet Layers	D (mm)	H (mm)	Number
1	C30N	C30	0	150	300	3
2	C30B6	C30	6	150	300	3
3	C30B12	C30	12	150	300	3
4	C30B18	C30	18	150	300	3
5	C30B24	C30	24	150	300	3
6	C30B30	C30	30	150	300	3
7	C50N	C50	0	150	300	3
8	C50B6	C50	6	150	300	3
9	C50B12	C50	12	150	300	3
10	C50B18	C50	18	150	300	3
11	C50B24	C50	24	150	300	3
12	C50B30	C50	30	150	300	3

**Table 2 polymers-13-04124-t002:** Compressive strength of concrete cubes.

Specimen	Compressive Strength (MPa)	Average (MPa)	Standard Deviation	Coefficient of Variation
C30-1	40.2	39.2	3.27	0.08
C30-2	42.5			
C30-3	34.7			
C50-1	73.7	70.6	2.20	0.03
C50-2	68.9			
C50-3	69.2			

**Table 3 polymers-13-04124-t003:** Summary of the test results of the specimens.

Specimen	Bamboo Sheet			*P_max_* (kN)	*f_cc_* (MPa)	*f_co_* (MPa)	*f_cu_* (MPa)	*ε_cc_*	*ε_cu_*
	*n_f_*	*t_f_* (mm)	*f_l_* (MPa)						
C30N-1	--	--	0	711	40.24	39.15	--	0.0025	--
C30N-2	--	--	0	750	42.48	39.15	--	0.0035	--
C30N-3	--	--	0	613	34.72	39.15	--	0.0020	--
C50N-1	--	--	0	1301	73.70	70.59	--	0.0030	--
C50N-2	--	--	0	1216	68.87	70.59	--	0.0031	--
C50N-3	--	--	0	1222	69.20	70.59	--	0.0030	--
C30B6-1	6	3.0	2.69	803.2	39.62	39.15	45.48	0.0023	0.0071
C30B6-2	6	3.0	2.69	824.7	39.29	39.15	46.7	0.002	0.0055
C30B6-3	6	3.0	2.69	838.9	40.92	39.15	47.53	0.0024	0.0070
C30B12-1	12	6.0	5.37	978.6	37.47	39.15	55.41	0.0017	0.0076
C30B12-2	12	6.0	5.37	1062.9	40.20	39.15	60.19	0.0023	0.0100
C30B12-3	12	6.0	5.37	952.3	37.95	39.15	53.92	0.0016	0.0070
C30B18-1	18	9.0	8.06	1145.7	44.05	39.15	64.87	0.0026	0.0075
C30B18-2	18	9.0	8.06	1048.5	41.08	39.15	59.39	0.0017	0.0067
C30B18-3	18	9.0	8.06	1238.3	46.01	39.15	70.12	0.0025	0.0093
C30B24-1	24	12.0	10.75	1146.8	46.72	39.15	59.27	0.0021	0.0051
C30B24-2	24	12.0	10.75	1069.0	46.63	39.15	60.53	0.0025	0.0060
C30B24-3	24	12.0	10.75	1074.6	47.95	39.15	60.85	0.0023	0.0057
C30B30-1	30	15.0	13.44	1149.6	46.20	39.15	65.10	0.0013	0.0051
C30B30-2	30	15.0	13.44	1104.1	49.71	39.15	62.51	0.0014	0.0053
C30B30-3	30	15.0	13.44	1077.2	48.56	39.15	61.00	0.0014	0.0057
C50B6-1	6	3.0	2.69	1215.0	68.03	70.59	68.80	0.0029	0.0042
C50B6-2	6	3.0	2.69	1182.0	59.80	70.59	66.93	0.0021	0.0040
C50B6-3	6	3.0	2.69	1083.1	57.83	70.59	61.33	0.0020	0.0026
C50B12-1	12	6.0	5.37	1206.4	68.31	70.59	68.31	0.0018	0.0018
C50B12-2	12	6.0	5.37	1263.4	71.54	70.59	71.54	0.0026	0.0021
C50B12-3	12	6.0	5.37	1141.0	63.10	70.59	64.60	0.0019	0.0030
C50B18-1	18	9.0	8.06	1325.7	72.26	70.59	75.06	0.0026	0.0035
C50B18-2	18	9.0	8.06	1373.1	52.53	70.59	77.75	0.0016	0.0040
C50B18-3	18	9.0	8.06	1404.0	74.87	70.59	79.50	0.0027	0.0036
C50B24-1	24	12.0	10.75	1348.3	76.35	70.59	76.35	0.0017	0.0017
C50B24-2	24	12.0	10.75	1343.5	76.07	70.59	76.07	0.0017	0.0017
C50B24-3	24	12.0	10.75	1236.0	69.99	70.59	69.99	0.0017	0.0017
C50B30-1	30	15.0	13.44	1107.3	66.77	70.59	66.77	0.0008	0.0008
C50B30-2	30	15.0	13.44	1385.8	74.46	70.59	78.47	0.0015	0.0037
C50B30-3	30	15.0	13.44	1315.6	74.49	70.59	74.49	0.0012	0.0012

**Table 4 polymers-13-04124-t004:** Strength models for FRP-confined concrete.

Models	Ultimate Strength
Samaan et al. (1998) [39]	$f_{cu}=f_{co}^{\prime }+6.0f_{l}{}^{0.7}$
Lam and Teng (2003) [40]	$\frac{f_{cu}}{f_{co}^{\prime }}=1+3.3\frac{f_{l,a}}{f_{co}^{\prime }}$
Wu et al. (2006) [41]	$\frac{f_{cu}}{f_{co}^{\prime }}=1+2.0\frac{f_{l}}{f_{co}^{\prime }}$ for strain-hardening $\frac{f_{cu}}{f_{co}^{\prime }}=0.75+2.5\frac{f_{l}}{f_{co}^{\prime }}$ for strain-softening
Youssef et al. (2007) [42]	$\frac{f_{cu}}{f_{co}^{\prime }}=1+2.25{\left(\frac{f_{l}}{f_{co}^{\prime }}\right)}^{1.25}$
Fahmy and Wu (2010) [43]	$f_{cu}=f_{co}^{\prime }+4.5f_{l}^{0.7}f_{co}^{\prime }\leq 40\mathrm{MPa}$ $f_{cu}=f_{co}^{\prime }+3.75f_{l}^{0.7}f_{co}^{\prime }>40\mathrm{MPa}$
Wei and Wu (2012) [44]	$\frac{f_{cu}}{f_{co}^{\prime }}=0.5+2.7{\left(\frac{f_{l}}{f_{co}^{\prime }}\right)}^{0.73}$
Wu and Wei (2014) [45]	$\frac{f_{cu}}{f_{co}^{\prime }}=0.75+2.7{\left(\frac{f_{l}}{f_{co}^{\prime }}\right)}^{0.9}$

Notes: *f_cu_* is the ultimate strength of confined concrete; $f_{co}^{\prime }$ is the unconfined concrete strength; *f_l_* is the lateral confining pressure; *f_l,a_* is the actual maximum confining pressure.

**Table 5 polymers-13-04124-t005:** Statistical assessment of the ultimate strength models.

Model	Average Value	Standard Deviation	Coefficient of Variation
Samaan et al. (1998) [39]	1.15	0.11	0.09
Lam and Teng (2003) [40]	1.00	0.10	0.10
Wu et al. (2006) [41]	1.01	0.11	0.10
Youssef et al. (2007) [42]	0.95	0.10	0.10
Fahmy and Wu (2010) [43]	1.03	0.08	0.08
Wei and Wu (2012) [44]	0.91	0.15	0.17
Wu and Wei (2014) [45]	0.96	0.13	0.14

## Data Availability

The data presented in this study are available on request from the corresponding author.

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
