# Peer review of "Compressive Behavior of Bamboo Sheet Twining Tube-Confined Concrete Columns"

_polymers, 2021, doi:10.3390/polym13234124_

Round 1

Reviewer 1 Report

The paper is aimed experimentally to study various axial compressive parameters of a new type of confined concrete, which is termed bamboo sheet twining tube-confined concrete. The paper is interesting and is in the journal scope. In my opinion this paper is acceptable but it needs clear revision:

(1) Keywords must indicate the main materials, tests, and methodology used in the study. Therefore, it is required to revise the keywords and write based on the points mentioned above.

(2) Please provide Research significance section and reduce Introduction section a bit. Please provide how this study fills the existing knowledge gap. What is the novelty of the manuscript ? It must be explicitly stated in the paper.

(3) It is very important to include more details about the experimental setup. What kind of equipment was used?  How was the load applied on the specimen? Why wasn’t it used by DIC to monitor the tests?

(4) For understanding of readers, please explain the meaning of each letter in all equations located in the paper.

(5) How was the compression loading applied? Pressure or displacement? Please provide more details.

(6) In the article the average values of compressive strength were given, e.g. in the line 97. What were the dispersions of the results? Please complete this.

(7) Figure 4, which presented differents types of specimens failure, should be better commented.

(8) References are suggestive. I am convinced that it is useful for the manuscript if will be included in the References section more recent papers with the same topics, or using similar procedures, i.e. articles on reinforcing concrete composites with novel materials - wchich were published earlier in mdpi journals, for example:

„Improvement of strength parameters of cement matrix with the addition of siliceous fly ash by using nanometric C-S-H seeds", Energies, 2020.

Author Response

Responses to the Reviewers’ comments

Thank you for your suggestions on this manuscript. We have addressed all of the comments in the discussion below, and the manuscript has been revised according to these suggestions. We believe that the manuscript will be more valuable and readable to the general audience after these revisions. The point-by-point responses to the comments made by the reviewers have been listed in detail below. In the revised manuscript, the modified text is highlighted for the convenience of reviewers.

Responses to Reviewer #1

General comment:

Reviewer #1:

The paper is aimed experimentally to study various axial compressive parameters of a new type of confined concrete, which is termed bamboo sheet twining tube-confined concrete. The paper is interesting and is in the journal scope. In my opinion this paper is acceptable but it needs clear revision:

Response to general comments:

Thanks for the positive comments and support of the reviewer. We have revised the manuscript carefully as recommended.

Comment 1:

Keywords must indicate the main materials, tests, and methodology used in the study. Therefore, it is required to revise the keywords and write based on the points mentioned above.

Response to comment 1:

Thank the reviewer for your good suggestions on the preciseness of the manuscript. The keywords have been revised (Keywords: Bamboo sheet; Bamboo composite tube; Confined concrete; Lateral-to-axial strain relationship; Ultimate strength).

Comment 2:

Please provide Research significance section and reduce Introduction section a bit. Please provide how this study fills the existing knowledge gap. What is the novelty of the manuscript? It must be explicitly stated in the paper.

Response to comment 2:

Thanks for your meaningful comment. Research significance section has been added from Line 50 to 70 and introduction section has been reduced a bit. The novelty of the manuscript has been stated from Line 73 to 81 in the introduction of the revised manuscript.

Comment 3:

It is very important to include more details about the experimental setup. What kind of equipment was used? How was the load applied on the specimen? Why wasn’t it used by DIC to monitor the tests?

Response to comment 3:

Thanks for your suggestion. More details about the experimental setup has been added from Line 137 to 144. The compression tests were conducted on a high-stiffness testing machine with a capacity of 3000 kN. Two control modes were conducted during the compression tests. Each specimen was first tested using load‐control at a loading rate of 10 kN/s until 10% of the pre-estimated peak load before formal loading to reduce the effect of looseness at both ends of the specimen. then, the load control was changed to displacement control at a constant rate of 0.1 mm/min until reaching specimen failure.

The measurement of strain distribution along specimen height and perimeter may be enabled by the digital image correlation (DIC) measurement technique. However, the data processing of DIC is complex. Installing strain gauges around the mid-height section and laser displacement sensors were adopted in this test, which have also been verified in most previous studies on FRP-confined concrete. (Line 146-150)

Comment 4:

For understanding of readers, please explain the meaning of each letter in all equations located in the paper.

Response to comment 4:

Thanks for your suggestion. The meanings of each letter in all equations located in the Table 4 have been added.

Comment 5:

How was the compression loading applied? Pressure or displacement? Please provide more details.

Response to comment 5:

Thanks for your suggestion. The compression tests were conducted on a high-stiffness testing machine with a capacity of 3000 kN. Two control modes were conducted during the compression tests. Each specimen was first tested using load‐control at a loading rate of 10 kN/s until 10% of the pre-estimated peak load before formal loading to reduce the effect of looseness at both ends of the specimen. then, the load control was changed to displacement control at a constant rate of 0.1 mm/min until reaching specimen failure. (Lines 137-144)

Comment 6:

In the article the average values of compressive strength were given, e.g. in the line 97. What were the dispersions of the results? Please complete this.

Response to comment 6:

Thanks for your suggestion. The average compressive strength of the two batches of concrete cubes were added in Table 2. The values of standard deviation (SD) and coefficient of variation (COV) is small, which means that the dispersions of the compressive strength are satisfactory. (Lines 104-106)

Comment 7:

Figure 4, which presented different types of specimen’s failure, should be better commented.

Response to comment 7:

The different types of specimen’s failure have been better commented from Line 177 to 193.

Comment 8:

References are suggestive. I am convinced that it is useful for the manuscript if will be included in the References section more recent papers with the same topics, or using similar procedures, i.e. articles on reinforcing concrete composites with novel materials - wchich were published earlier in mdpi journals, for example: Improvement of strength parameters of cement matrix with the addition of siliceous fly ash by using nanometric C-S-H seeds", Energies, 2020.

Response to comment 8:

Thanks for your meaningful suggestion. More recent relative references have been added in the introduction of the revised manuscript. (Line 33)

Reviewer 2 Report

This manuscript needs essential modifications as follows:

  • The introduction should be revised. Group citations in one phrase should be avoided; see Page 3, Line 16. All irrelevant references must be omitted (most of the first nineteen Refs.). Updated and relevant references must be added and discussed, see for example:
    • https://doi.org/10.1007/s00107-021-01737-8
  • A comparison with the above Ref. must be made, and subsequently, the novelty of the present work should be explained well.
  • The assumptions and Ref. of Eq. 1 must be added.
  • Line 255, "The results from the Ozbakkaloglu et al. model ---". The Ref. must be added.
  • 4.2 must be extended, "the models proposed by Wu and Wei and Wei and Wu" is only mentioned in conclusion # 5. The References number must be stated in Figs. 9 & 10, and Table 4.

Author Response

Responses to the Reviewers’ comments

Thank you for your suggestions on this manuscript. We have addressed all of the comments in the discussion below, and the manuscript has been revised according to these suggestions. We believe that the manuscript will be more valuable and readable to the general audience after these revisions. The point-by-point responses to the comments made by the reviewers have been listed in detail below. In the revised manuscript, the modified text is highlighted for the convenience of reviewers.

Responses to Reviewer #2

General comment:

Reviewer #2:

This manuscript needs essential modifications as follows:

Response to general comment:

Thank you for your time and patience. We have revised the manuscript carefully as recommended.

Comment 1:

The introduction should be revised. Group citations in one phrase should be avoided; see Page 3, Line 16. All irrelevant references must be omitted (most of the first nineteen Refs.). Updated and relevant references must be added and discussed, see for example: https://doi.org/10.1007/s00107-021-01737-8.

Response to comment 1:

Thank you for your suggestion. The introduction has been revised in the introduction of the revised manuscript (Lines 31-81). Group citations in one phrase have been avoided (Line 41 and 43). All irrelevant references have been omitted. Relevant references have been added and discussed. (Lines 66-70)

Comment 2:

A comparison with the above Ref. must be made, and subsequently, the novelty of the present work should be explained well.

Response to comment 2:

Thanks for your meaningful comment. The comparison with the above Ref. and the novelty of the present work have been added in the introduction of the revised manuscript. (Lines 75-81).

Comment 3

The assumptions and Ref. of Eq. 1 must be added.

Response to comment 3:

The assumptions and Ref. of Eq. 1 has been added in Line 168-171 of the revised manuscript.

Comment 4

Line 255, "The results from the Ozbakkaloglu et al. model ---". The Ref. must be added.

Response to comment 4:

The Ref. has been added in Line 279.

Comment 5

4.2 must be extended, "the models proposed by Wu and Wei and Wei and Wu" is only mentioned in conclusion # 5. The References number must be stated in Figs. 9 & 10, and Table 4.

Response to comment 5:

Thanks for your suggestion. "the models proposed by Wu and Wei and Wei and Wu" has been added in Line 312-320. And the References number has been stated in Figs. 9 & 10, and Table 4.

Round 2

Reviewer 1 Report

I have no comments.

Author Response

Thank you for your time and patience.

Reviewer 2 Report

The authors have successfully addressed most of my comments.  Therefore, I recommend the publication of this manuscript after correcting this minor error:

  • The authors substituted "Ozbakkaloglu et al." with "Lim and Ozbakkaloglu", and they did not delete "et al.", which follows "Ozbakkaloglu". Therefore, "et al." must be deleted. This comment applies to the entire manuscript.

Author Response

Responses to Reviewer #2

General comment:

Reviewer #2:

The authors have successfully addressed most of my comments. Therefore, I recommend the publication of this manuscript after correcting this minor error:

Comment 1:

The authors substituted "Ozbakkaloglu et al." with "Lim and Ozbakkaloglu", and they did not delete "et al.", which follows "Ozbakkaloglu". Therefore, "et al." must be deleted. This comment applies to the entire manuscript.

Response to comment 1:

Thank you for your meaningful suggestions on this manuscript. We have revised the entire manuscript carefully as recommended.
